# Reduced Vessel Density and Enlarged Foveal Avascular Zone in the Macula as a Result of Systemic Hypoxia Caused by SARS-CoV-2 Infection

**DOI:** 10.3390/jpm13060926

**Published:** 2023-05-31

**Authors:** Magdalena Kal, Bernadetta Płatkowska-Adamska, Dorota Zarębska-Michaluk, Piotr Rzymski

**Affiliations:** 1Collegium Medicum, Jan Kochanowski University, 25-317 Kielce, Poland; kalmagda@gmail.com (M.K.); dorota1010@tlen.pl (D.Z.-M.); 2Ophthalmic Clinic, Voivodeship Hospital, 25-736 Kielce, Poland; 3Department of Infectious Disease, Provincial Hospital, 25-317 Kielce, Poland; 4Department of Environmental Medicine, Poznań University of Medical Sciences, 60-806 Poznan, Poland; rzymskipiotr@ump.edu.pl

**Keywords:** COVID-19, SARS-CoV-2, optical coherence tomography, retinal and choroidal microvasculature

## Abstract

Infection with SARS-CoV-2 can lead to various long-term consequences, including those of an ophthalmic nature. This paper reviews the results of optical coherence tomography angiography (OCTA) performed among COVID-19 patients. The review included papers evaluating short- and long-term outcomes following the SARS-CoV-2 infection. Some differentiated the obtained retinal and choroidal vascularization parameters according to gender. Following COVID-19, patients reveal changes in retinal and choroidal vascular parameters based on OCTA, such as reduced vascular density and an increased foveal avascular zone, which can persist for several months. Routine ophthalmic follow-up with OCTA should be considered in patients after SARS-CoV-2 infection to assess the effects of inflammation and systemic hypoxia in COVID-19. Further research is needed to understand whether infection with particular viral variants/subvariants may vary in the risk of effects on retinal and choroidal vascularization and whether and to what extent these risks may also differ in relation to reinfected and vaccinated individuals.

## 1. Introduction

The clinical spectrum of infection with severe acute respiratory syndrome coronavirus 2 (SARS-CoV-2) is vast, including predominantly asymptomatic and mild cases as well as a severe course with progressive pneumonia and acute respiratory distress syndrome, both of which can be accompanied by cytokine storm, thrombosis, and multiple organ dysfunction [1,2,3,4]. Moreover, a wide range of persistent symptoms can remain after SARS-CoV-2 infection, known as post-COVID-19 syndrome (post-coronavirus disease 2019), which can encompass sensory, neurologic, cardiorespiratory, and mental health [5].

SARS-CoV-2 can also potentially lead to ophthalmic complications, including conjunctivitis, retinovascular disease, uveitis, optic neuropathies, and orbital fungal co-infections [6,7,8]. These can be an indirect effect of proinflammatory agents triggered in response to the viral infection or due to the direct interaction of SARS-CoV-2 with angiotensin-converting enzyme 2 (ACE2), which is utilized as a cellular entry receptor. ACE2 is ubiquitously present on the surface of various cell types, including type II alveolar cells, small intestinal enterocytes, arterial and venous endothelium, and arterial smooth muscle of most organs. It can also be found in the choroid of the eye and various retinal cells such as Muller cells, ganglion cells, retinal vascular endothelial cells, and photoreceptors [9,10,11]. Importantly, various co-receptors that facilitate cleavage of the SARS-CoV-2 spike protein following interaction with ACE2, i.e., type I transmembrane serine-protease furin and transmembrane protease 2 (TMPRSS2), are present in various locations of the ocular system [9,10,11].

Moreover, SARS-CoV-2 can alternatively utilize neuropilin 1 for cellular entry [12,13]. It is expressed by endothelial cells and retinal pigment epithelial cells located in choroidal neovascular membranes [14,15]. It also plays a role as a co-receptor for vascular endothelial growth factor, which is pivotal for ocular angiogenesis [16]. All in all, neuropilin 1 is involved in retinal and choroidal neovascularization [17]. Hence, its interaction with SARS-CoV-2 can lead to adverse interference, resulting in ocular manifestations and further consequences for COVID-19 convalescent patients. In conclusion, it is plausible that SARS-CoV-2 infection could lead to ocular manifestations, as already reviewed elsewhere [17], but also initiate adverse functional alterations in the retinal and choroidal vasculature of the eye of infected patients.

There are different methods to analyze ocular vascularity and circulation [18,19]. OCT is a non-invasive method that allows for the assessment of retinal and choroidal microvasculature. The use of this technique, access to which has increased significantly over the years, allows obtaining parameters such as vessel density (VD) and foveal avascular zone (FAZ) in various systemic conditions such as carotid stenosis, anemia, and COVID-19 [19,20,21,22].

The aim of the present study was to evaluate the effect of the SARS-CoV-2 infection and the following hypoxia on the vascularization of the retina and choroid of the human eye based on the optical coherence tomography angiography (OCTA) study, which objectively presents numerical parameters assessing the VD of the above structures of the posterior segment of the eye. To this end, a Boolean search of the PubMed/Medline database was performed to retrieve records and abstracts of English-language articles published in peer-reviewed journals. To this end, the following combination of keywords was employed: ”vessel density”, “retinal and choroidal microvasculature”, and “optical coherence tomography angiography”. This allowed the selection of the original research articles and case reports published between January 2020 and March 2022 that report changes in retinal and choroidal microvasculature in individuals who underwent SARS-CoV-2 infection. Following the literature review, the present paper was divided into the following subsections to detail the relevant parameters analyzed in COVID-19 patients based on OCTA: VD, FAZ, changes in OCTA parameters fixed over time, and gender analysis.

## 2. Main Characteristics of Optical Coherence Tomography

OCT plays an important role in the detailed diagnosis of the anterior and posterior structures of the eye. An extension of this tool is OCTA, which allows the assessment of the microcirculation in the retina and choroid during ocular and systemic diseases. Both methods are high-resolution diagnostic tools that enable rapid and non-invasive examinations [23]. SS-OCT (swept-source OCT) is the most modern version of OCT, incorporating a tunable light source. The light source is characterized by a narrow spectrum, and the wavelength frequency varies over time within a specific range. The most significant advantage of this method is its current highest scanning speed of 370,000 scans A/s [24]. The light wavelength is 1050 nm (infrared), the axial resolution is 2.6 µm, the lateral resolution is 14 µm^2^, and the scanning width is 12 mm. Penetration of the light waves extends to the sclera and allows evaluation of deep layers of the eyeball, such as the choroid. The resulting image of the individual layers of the retina and choroid corresponds to the histological structure of these tissues. Using SS-OCT technology, automatic thickness and volume maps of the retinal and choroidal layers are created. On the other hand, OCTA enables the diagnosis of pathological changes in the choroid and retina in various ocular and systemic vascular diseases [25].

The effect of OCTA is based on a split-spectrum amplitude decorrelation algorithm that amplifies the vascular signal while reducing background noise. This involves the acquisition of consecutive B-scans of the same tissue section, allowing the detection of erythrocyte movement. Cells that are in the same place at the time of examination appear identically in successive B-scans of the examined area. Changes in the examined cross-section indicate areas of erythrocyte movement in the vessel lumen. By detecting these moving erythrocytes in all areas examined, the algorithm creates a map of the retinal and choroidal blood vessels [26].

OCTA images must be interpreted correctly and cautiously, as artifacts may sometimes occur and limit the analysis of the results. These artifacts are due to the structure of the eye, eye movements, and ocular pathology and sometimes arise due to the limitations of the software of instruments containing this examination technique. In general, they can be classified into flow and motion projection artifacts. The software in state-of-the-art OCTA instruments reduces these artifacts. The sizes of the scans analyzed are typically 3 × 3 mm, 6 × 6 mm, and 12 × 12 mm. Larger scans have lower resolution [27].

The following OCTA parameters are automatically assessed during the analysis: VD in three different plexuses: superficial capillary plexus (SCP), deep capillary plexus (DCP), and choriocapillaris (CC) using EDTRS (Early Treatment of Diabetic Retinopathy Study) grid subfields to define areas of interest. VD is measured in the foveal (F), superior (S), inner nasal (N), inner inferior (I), and inner temporal (T) areas. In SCP and DCP, a foveal avascular zone (FAZ) can be manually delineated, encompassing the central part of the fovea, where no clear and demarcated vessels can be seen on OCTA [28] (Figure 1).

## 3. Retinal Ischemia

The inner retina, starting from the outer plexiform layer, consists of two nuclear layers, two plexiform layers, and a layer of nerve fibers and is supplied by the central retinal artery [29]. It is important to emphasize the vasculature’s role in nourishing the retina’s outer layers, the retinal pigment epithelium, and parts of the optic nerve. This is the only source of nutrition in the fovea, which consists entirely of photoreceptors and Muller cells. The choroid, as one of the most vascularized tissues in the human body, is responsible for the oxygenation of the outer retina [30]. The vascular endothelium plays an important role in the retinal vasculature. It controls the vascular tone and is responsible for the integrity of the blood-retinal barrier. Mechanisms such as vasospasm or inflammation can lead to vascular endothelial dysfunction and subsequent retinal ischemia [31].

The presence of SARS-CoV-2 viral particles was confirmed in the results of retinal biopsies from enucleated eyes among patients who had severe pulmonary COVID-19, had been hospitalized in the intensive care unit, required mechanical ventilation, and eventually died. By employing transmission electron microscopy, viral particles were visualized in the perinuclear region of cells and capillary endothelial cells of the inner nuclear layer and in the perinuclear and cytoplasmic regions of the outer nuclear layer. Immunofluorescence microscopy confirmed the presence of spike protein and nucleocapsid protein in the retina [32].

Patients hospitalized with COVID-19 also had retinal and choroidal abnormalities manifested by fundus and OCT examinations. In addition, RNA from SARS-CoV-2 has been detected in tears and conjunctival secretions from selected patients, indicating the potential ocular tropism of SARS-CoV-2 [33].

Marinho et al. found the presence of cotton wool spots and microhemorrhages along the retinal vessels in adult patients with COVID-19 11–33 days after the first symptoms of the viral disease. All these patients had fever, weakness, and dyspnea, and 11 had no sense of smell. Two patients were admitted to the hospital; the rest were treated on an outpatient basis. No patient required intensive care. All patients had normal blood parameters on ophthalmological examination. These researchers demonstrated the presence of hyperreflective changes in the retinal ganglion cell layer (GCL) and inner plexiform layer (IPL) on OCT [34]. Other authors have also described microhemorrhages, cotton wool spots, and tortuosity in hospitalized patients with acute COVID-19 [19,20]. Pereira et al. noted flame-shaped hemorrhages and cotton wool spots on fundus examination in hospitalized patients with severe COVID-19 [35]. The exact physiopathology of cotton wool spots after COVID-19 is not known. Probably, the vascular obstruction caused by direct viral tissue injury and infiltration of the endothelial cells via the ACE2 receptor can lead to this symptom [36].

There are also case reports of COVID-19 patients diagnosed with fundus lesions. One study described a 46-year-old man with no additional diseases and saturation on the day of hospital admission of 93%. Three days after initial respiratory symptoms, a fundus examination showed cotton-wool spots and tortuous retinal vessels. OCT examination showed a thickening of the inner layers of the retina. Three months later, all vascular changes present on the fundus had resolved. The visual acuity was normal during both ocular examinations [37].

Both central retinal vein and artery occlusions have been reported in patients treated for COVID-19 who have no typical systemic vascular risk factors. Plausibly, the complement-induced prothrombotic and inflammatory state induced by the virus is responsible for endothelial damage and microangiopathic injury [38]. Yahalomi et al. described a previously healthy 33-year-old female with central retinal vein occlusion with COVID-19 [39]. Invernizzi et al. found retinal hemorrhages, cotton wool spots, dilated veins, and tortuous vessels in 54 COVID-19 patients in fundus examination. As suggested, retinal vein diameter correlated directly with disease severity. This symptom can be a non-invasive parameter to monitor the inflammatory response and endothelial injury caused by SARS-CoV-2 infection [40].

Hommer et al. studied the effect of hypoxia in 24 healthy volunteers (mean age 26 years) on retinal vascularization parameters based on OCTA during breathing a mixture of 88% nitrogen and 12% oxygen [41]. Perfusion density in the superficial vascular plexus (SVP) increased significantly from 34.4 +/− 3.0 a.u. to 37.1 +/− 2.2 a.u., and remained stable in the DCP. There was a significant increase in retinal vessel diameter. During 100% oxygen breathing, which meant that study participants were put into a hyperoxic state, a significant decrease in DCP perfusion density was observed from 41.7 +/− 2.4 a.u. to 35.6 +/− 3.1 a.u., which was accompanied by a reduction in vessel diameter in the major retinal arteries and veins. An appropriate oxygen supply is required for the retina to function properly. Retinal circulation can adjust blood flow in response to both hypoxia and hyperoxia. A reduction in the caliber of the retinal vasculature in response to hyperoxia is a counter-regulatory mechanism to avoid excessive oxygen delivery to the retina, which can be toxic to neurons. The vasoconstriction response in DCP reduces the risk of hyperoxygenation and oxidative stress [41]. The exact reasons behind an increase in vascular density in the SCP in response to hypoxia are currently unknown. Even slight hypoxia alters the oxygen gradient between the choroidal capillaries and the outer retina, which slows the metabolic activity of the photoreceptors. In contrast, the retinal ganglion cell layer is more sensitive to hypoxia, which can induce an autoregulatory response mainly in the superficial choroid plexus [41].

## 4. Reduced Vessel Density in Patients with COVID-19

Many studies have reported changes in retinal VD among patients with COVID-19 based on OCT [42,43,44,45,46,47,48]. One study reported a reduced VD in SCP in patients with acute COVID-19 (*n* = 25, mean age 61 years, 14 days after hospital discharge) in the parafoveal area and foveal area in the macula [42]. Another study described no statistically significant differences in VD between patients with COVID-19 (mean age 50 years, 32 inpatients and 46 outpatients 1–4 months after positive PCR result) and healthy controls in the foveal area and in the parafoveal superior, inferior, nasal, and temporal areas in SCP [43]. Abrishami et al. found lower VD in SCP and DCP in the central retina among patients examined two weeks after recovery from acute COVID-19 (*n* = 31, mean age 40 years). Direct viral infection of the retina and a secondary phenomenon of systemic inflammation are likely responsible for this condition [44]. Other investigations reported significantly lower VD in SCP in patients with COVID-19 (*n* = 66, mean age 57, 12 weeks after diagnosis) compared to healthy controls in the central area, inner ring, and outer ring [45]. In another study, significantly lower VD was observed in COVID-19 patients (*n* = 50, mean age 37 years), during the unspecified period after COVID-19 treatment, in the hemi quadrant, upper quadrant, and lower quadrant [46]. Other researchers found no significantly reduced VD was observed in the foveal and parafoveal areas of the macula in COVID-19 patients in SCP (*n* = 63, mean age 51 years, 2 months after hospital discharge) compared to healthy controls [47].

Another parameter analyzed in COVID-19 patients was VD in DCP. One study demonstrated a significantly lower VD in patients affected by COVID-19 in the parafoveal, superior hemi, and superior quadrant of the DCP [46]. In turn, in the study by Kal et al., no significantly reduced VD was observed in the foveal and parafoveal areas of the macula in DCP compared to healthy controls [47].

A few papers have analyzed VD in CC [43,47]. Some researchers found significantly lower VD in the foveal area of the CC compared to the healthy group, but significantly reduced VD in the parafoveal areas of this plexus compared to the healthy group was not present [47]. Another study noted no statistically significant reduction in VD of the foveal area and parafoveal CC [42]. The factor that may have influenced the differences in parameters based on OCTA obtained between patients with COVID-19 and healthy subjects was the timing of the eye examination.

As explained, the resistance of the choroidal circulation does not change according to the metabolic demand of the outer retina. Hypoxia leads to an increase in arterial pressure, which in turn causes an increase in choroidal blood flow. Retinal blood flow increases with a decrease in oxygen partial pressure, while increased choroidal blood flow is not observed [49].

In another study analyzing VD in COVID-19, the patients were divided according to the severity of SARS-CoV-2 infection: (i) mild disease (*n* = 24, mean age 41, outpatients); (ii) moderate disease, patients with COVID-19 pneumonia with interleukin-6 levels below 40 pg/mL (*n* = 24, mean age 42, 72 days after leaving hospital); and (iii) severe disease, patients with COVID-19 pneumonia with interleukin-6 levels above 40 pg/mL (*n* = 21, mean age 44, 70 days after leaving hospital). Compared to mild and symptomatic patients and the healthy, uninfected control group, foveal VD in SCP was lower in patients with moderate and severe COVID-19 [48].

The above studies show discrepancies in statistical significance for the VD parameter in COVID-19 patients. This may be due to the small study groups or to differences in the time of the first eye examination from the end of hospitalization. Some were examined for the first time after 14 days, while others were examined 12 weeks after leaving the hospital [42,43,46,47]. There is no precise data on the duration of hospitalization for these patients. In most cases, patients were not examined ophthalmologically during their hospitalization due to the risk of staff being infected with SARS-CoV-2.

## 5. The Enlarged Foveal Avascular Zone in Patients with COVID-19

The FAZ area is characterized by a high density of choriocapillaris and a lack of blood vessels, so the choriocapillaris supply all the oxygen and nutrients to the area [50]. The FAZ is very sensitive to ischemia and hypoxia, so its enlargement can be present in many vascular pathologies, such as diabetic retinopathy and retinal vein occlusion. Therefore, the FAZ can be expected to be enlarged after COVID-19.

One study found a statistically larger FAZ area in the SCP in patients with COVID-19 than in controls. However, oxygen saturation (OS) values were not reported [42]. Another investigation demonstrated no statistically significant difference in FAZ size between SCP and DCP in COVID-19 convalescent patients. In this group, oxygen was supplemented for 26 patients, while 46 patients were not hospitalized. These authors did not report saturation values in the study group [43]. Similar observations were made by Guemes-Villahoz et al., who did not find a significantly larger FAZ area in SCP in patients after COVID-19. The mean OS in this group was 96%, but six patients (8%) had OS < 92% [45]. Leyla Hazar et al. also found no difference between patients and controls in FAZ size in SCP and DCP, while all patients had OS > 90% [46].

A study by Kal et al. observed a significantly increased FAZ area in DCP in patients hospitalized for COVID-19 compared to controls. In this studied group, twenty-two patients were classified at baseline as stable with oxygen saturation (OS) > 95%, twenty-nine were unstable with OS 91–95%, and the remaining twelve patients were assessed as unstable with OS ≤ 90% [47]. Zapata et al. also reported an enlarged FAZ area in patients with moderate and severe COVID-19 when compared to a mild infection and a healthy, uninfected control [48]. However, this contrasts with another study in which the FAZ area was not significantly larger in DCPs in COVID-19 patients [42].

Nevertheless, the overall conclusion is that FAZ can be enlarged in COVID-19 patients, indicating that it might be induced by hypoxia. Differences in the FAZ parameter between COVID-19 patients and control groups may be explained by factors like those that may impinge on the VD parameter mentioned above. Perhaps the patients’ age, the burden of additional diseases, saturation, treatment during hospitalization, and the time of the first ocular examination after hospital discharge. Retinal ischemia, due to endothelial dysfunction, vasospasm, and hypercoagulability, may lead to enlargement of the FAZ area. Systemic hypoxia and inflammation in patients with COVID-19-related pneumonia can cause a perfusion deficit in the macular area of the retina [32,51]

## 6. Persistent Reduced VD and Enlarged FAZ at 6-Months Follow-Up

More than half of patients develop a symptom syndrome termed “long-COVID” after COVID-19, which is defined as the continuation or development of new symptoms 3 months after the initial SARS-CoV-2 infection, with these symptoms lasting for at least 2 months with no other explanation [52]. Over 200 symptoms have been associated with long COVID-19, with fatigue being the most common [53,54]. However, it can also affect sensory, neurologic, and cardiorespiratory systems and mental health [55]. Therefore, it is likely that it is a multifaceted phenomenon that may arise from immune dysregulation with or without reactivation of herpesviruses, persistent SARS-CoV-2 infection in certain locations (e.g., gastrointestinal tract), microbiota disruption, autoimmunity, clotting and endothelial abnormalities, and dysfunctional neurological signaling [54,56,57,58,59]. According to some estimates, long COVID-19 has a substantial prevalence of 43% and more frequently affects hospitalized patients [60]. Moreover, it appears to be more common in women than in men [38].

Long COVID-19 may also have ocular manifestations [61,62,63,64,65]. One study included patients with COVID-19 (*n* = 25, mean age 40 years) in the early period after hospital discharge and six months later. These patients did not require a stay in the intensive care unit. The VD in CC was reduced in the second evaluation, carried out after six months. However, no differences in VD in the SCP and DCP plexus were observed [66]. When patients, divided according to COVID-19 severity into mild, moderate, and severe groups, were re-examined eight months after their first ophthalmological examination, performed soon after leaving the hospital, it was found that severely infected individuals tended to exhibit a larger FAZ area but no significant differences in VD values [67]. In a study conducted by Kal et al., significantly decreased VD in the SCP was described among patients who recovered from COVID-19, and the persistence of these changes for six months was noted in the foveal area of the SCP and the superior, nasal, and inferior areas of the SCP [54,68]. None of the patients in this study required a stay in the intensive care unit; 26 patients of the 63 hospitalized had confirmed bilateral pneumonia by computerized tomography scans. The persistence of significantly decreased VD in DCP in COVID-19 patients was demonstrated at the 6-month examination (mean age 51 years) compared to baseline evaluation in the foveal area of DCP, superior, nasal, and inferior areas of DCP. At the same time, it significantly increased VD in the CC foveal area but significantly decreased VD in the CC superior, nasal, inferior, and temporal areas. The FAZ area also remained enlarged in DCP during the 6-month follow-up of the same group of patients [68]. Changes in retinal and choroidal microvascular parameters on OCTA were still observed six months later. Similar results are found in the observation of the brain, which is embryologically and structurally like the retina. One study reported persistent abnormalities in cerebral blood flow three months after the recovery of patients hospitalized for COVID-19 pneumonia [69].

## 7. Gender-Related Persistent Reduced VD and Enlarged FAZ Area in Patients with COVID-19

Several papers present differences in the course of COVID-19 according to gender. Males experience a higher severity, hospitalization rate, and fatality rate for COVID-19 infection than females [70,71,72,73]. These differences are likely to arise from differences in genetic factors (e.g., chromosome X inactivation in the immune response to SARS-CoV-2), ACE2 expression, immune function, and endocrine regulation [74,75,76,77,78,79]. Expression of the ACE2 gene is regulated by estrogens, while that of TMPRSS2 is controlled by an androgen-responsive promoter. Higher levels of ACE2 in women provide a protective effect against lung damage [79]. By promoting innate and acquired immune responses, estrogens facilitate the clearance of pathogens from the female body. On the other hand, testosterone inhibits immune function and counteracts estrogen-influenced pathways, resulting in increased susceptibility to infectious diseases in men [80]. Moreover, women have a higher number and activity of innate immune cells, including monocytes, macrophages, and dendritic cells, compared to men, resulting in faster intervention against pathogens. The more robust inflammatory response in women, even at older ages, during SARS-CoV-2 infection is due to the gender-specific activation of T lymphocytes early in this viral infection [81,82].

Bilbao-Malave observed patients (*n* = 17, mean age 61) who recovered from COVID-19 two weeks after discharge and six months later, finding reduced VD in the central area of the SCP and DCP and a widened FAZ compared to healthy patients. Women showed a statistically significantly reduced VD in the foveal area of SCP over a period of 6 months, which was not found in the male group. The FAZ area was also increased in the second follow-up study in both the female and male groups [83].

In another study, researchers observed differences in VD between males and females within groups that suffered from mild, moderate, and severe COVID-19 eight months after the first ocular examination. The central VD in SCP at baseline for the mild COVID-19 group was higher in women than in men. In turn, in the moderate COVID-19 group, women revealed higher inferior VD in DCP at 8-month follow-up, while men exhibited higher inferior VD in SCP and central VD in DCP [67].

In the study by Kal et al., women had a statistically significantly reduced VD foveal in SCP at the first visit. The FAZ area in SCP was also significantly increased in women compared to men [68] (Figure 2). DCP analysis also revealed gender-related differences: VD was significantly reduced in women in the temporal area at the first visit, which was not found in the male group. Six months later, VD in the frontal area, like VD in the temporal area, was reduced in women compared to men. In CC, the female group presented increased VD in the supratemporal area 6 months after hospitalization [68].

Several investigators find a gradual deterioration of ocular parameters based on OCTA in women compared to men after COVID-19 and a slower recovery process in women after SARS-CoV-2 infection. Consequently, a higher risk of decreasing VD in women may be observed over several months after COVID-19 [67,68].

Sex hormones also influence the regulation of ocular vascularization. Estrogen is a protective hormone in terms of decreased vascular resistance in the large vessels of the eye. One study found greater choroidal blood flow in women under 40 compared to women over 55; in men, age did not affect choroidal blood flow through the eyeball. These results may depend on hormonal management in both groups [84]. Further observations should be made among men and women regarding differences in ocular vascularization based on the OCTA study to draw further conclusions.

## 8. Future Research Prospects

Whether the effect of SARS-CoV-2 infection on the microcirculation of the retina and choroid could differ between different viral variants is not well explored. One should note that such a phenomenon may be expected since there are significant differences in the clinical course of COVID-19 between particular lineages, with a highly transmissible Omicron variant recognized as the least severe [85,86,87,88]. Moreover, particular subvariants of Omicron reveal differences in fusogenicity, which is known to affect viral pathogenicity [89]. The potential effects of infection may also vary in relation to viral loads [90,91]. Whether vaccination status may attenuate the effect of SARS-CoV-2 on the retinal and choroidal vasculature of the eye during breakthrough infection also remains to be explored.

In the case of abnormalities in the morphology and vascularization of the retina and choroid in the macula diagnosed with OCT in patients after COVID-19, one may consider expanding the diagnosis with invasive fluorescein angiography. This test can diagnose the posterior segment of the eye and confirm the reliability of the utility of OCT. It is worth considering regular ocular diagnosis with the above diagnostic tests on a regular basis, especially in patients with severe COVID-19.

## 9. Conclusions

According to literature data, FAZ enlargement and VD reduction in SCP and DCP based on OCTA have been documented in patients who underwent COVID-19, and these changes persist for at least six months of follow-up. OCTA helps objectively assess the microcirculation of the retina and choroid, and its parameters can be considered biomarkers of vascular damage in other organs due to SARS-CoV-2 infection. Due to the history of COVID-19, a general medical history of SARS-CoV-2 infection should be considered in differentiating macular disease based on OCTA examination. Further follow-up studies are needed in this group of patients stratified by gender to assess whether there are consequences for these ocular structures due to impaired blood supply.

## Figures and Tables

**Figure 1 jpm-13-00926-f001:**
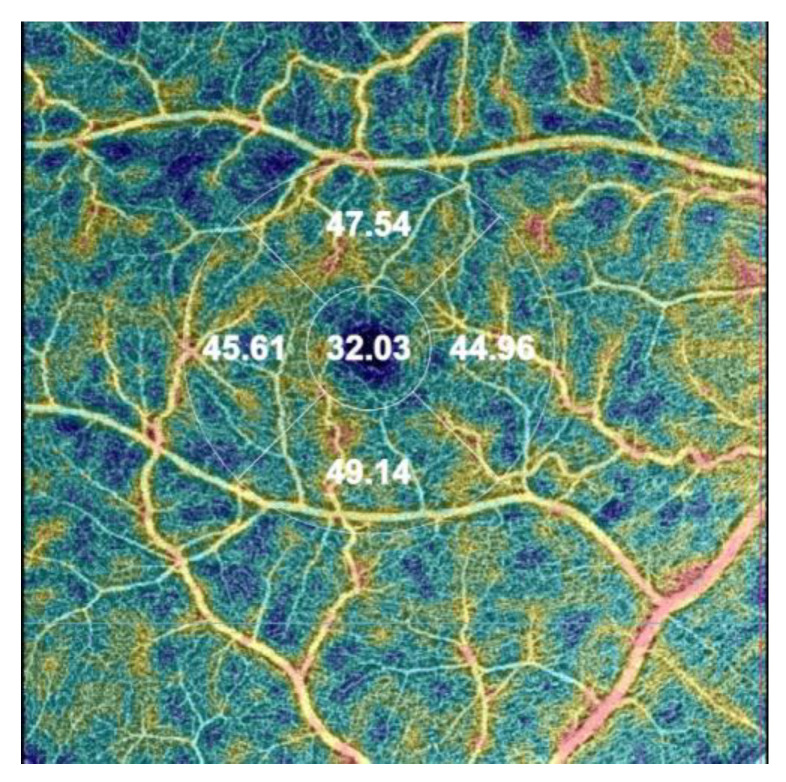
The image of the vessel density (VD) map in the right eye was assessed automatically by optical coherence tomography angiography (OCTA) using the early treatment diabetic retinopathy (ETDRS) grid situated in the fovea by fixation in the superficial capillary plexus (SCP). The automatic map of central VD is divided into four areas: the foveal VD (F VD = 32.03%), the superior area (S VD = 47.54%), the inferior VD (I VD = 49.14%), the temporal VD (T VD = 45.61), and the nasal VD (N VD = 44,96%).

**Figure 2 jpm-13-00926-f002:**
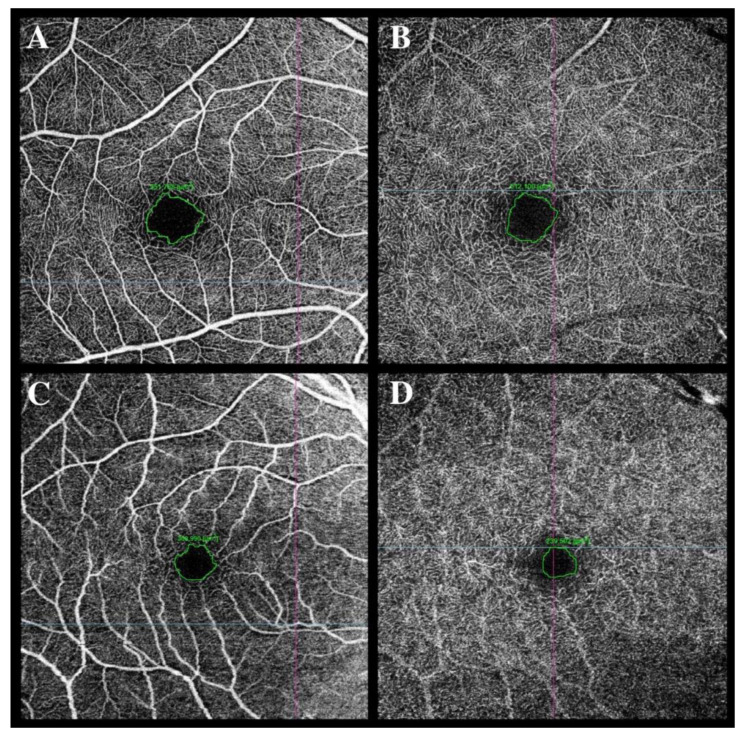
Foveal avascular zone (FAZ) enlargement was observed in a female patient in the superficial capillary plexus (SCP) (**C**) and the deep capillary plexus (DCP) (**D**) at 2 months after hospital discharge compared to the FAZ area in SCP (**A**) and DCP (**B**) in a male patient at 2 months after hospital discharge (**A**,**B**).

## Data Availability

All data is included in the published paper.

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
