# Peer review of "Reduced Vessel Density and Enlarged Foveal Avascular Zone in the Macula as a Result of Systemic Hypoxia Caused by SARS-CoV-2 Infection"

_jpm, 2023, doi:10.3390/jpm13060926_

Round 1

Reviewer 1 Report

The manuscript titled “The reduced vessel density and enlarged foveal avascular zone in the macula as a result of systemic hypoxia caused by SARS-CoV-2 infection “is a good attempt. However I feel there is a lack of little information and I have concerns about some of the following queries to be clarified before publication. 

Comments 

1. Introduction part is lacking the functional details of target, for that, the authors are requested to provide the detailed functional importance of target.

2. In title of the paper authors have mentioned that impact of COVID-19 on ophthalmic nature with results of optical coherence tomography angiography, but in the main text it was discussed about the general working principle of the OCTA and other methods, no proper discussion on the impact of ophthalmic nature and results of OCTA in post-COVID-19 outpatients. 

3. Authors have mentioned that the results of the biopsies from enucleated eyes among patients who died COVID-19 shows the expression of spike protein and nucleocapsid  in the retina. Is this expression found only in the severely infected patients or is it also predicted in mild to moderate infected people?

4. The presence of cotton wool spots and microhemorrhages along the retinal vessels in adult patients with COVID-19 is present in all the types of patients or severely infected patients only? Provide this information more in detail. 

5. The study reports on reduced vessel density in patients shows both significant and no statistically significant reduction in VD in CC, and authors have reported that a number of factors may influence the difference in parameters, if authors discuss these factors more in detail it could be more informative.

6. Authors have mentioned that hypoxia is responsible for the increased FAZ in DCP in patients, but also it was reported that no difference between  

7. Authors have mentioned that the changes in retinal and choroidal microvascular parameters on OCTA were still 230 found 6 months later, is this report found in only severely infected patients, or in both mild and moderate infection?

Minor edits required

Author Response

Reviewer 1

  1. Introduction part is lacking the functional details of target, for that, the authors are requested to provide the detailed functional importance of target.

Answer to the review, not included in the text:

As suggested, we have changed the introduction.

Additional information included in the manuscript:

Section: Introduction; P. 2 of 14; line 79-83:

The aim of the present study was to evaluate the effect of the SARS-CoV-2 infection and the following hypoxia on the vascularisation of the retina and choroid of the human eye based on the optical coherence tomography angiography (OCTA) study, which objectively presents numerical parameters assessing the VD of the above structures of the posterior segment of the eye.

  1. In title of the paper authors have mentioned that impact of COVID-19 on ophthalmic nature with results of optical coherence tomography angiography, but in the main text it was discussed about the general working principle of the OCTA and other methods, no proper discussion on the impact of ophthalmic nature and results of OCTA in post-COVID-19 outpatients. 

Answer to the review, not included in the text:

We also refer to a publication that describes post-COVID-19 outpatients: (Szkodny, D.; Wylęgała, E.; Sujka-Franczak, P.; Chlasta-Twardzik, E.; Fiolka, R.; Tomczyk, T.; Wylęgała, A. Retinal OCT Findings in Patients after COVID Infection. J. Clin. Med. 2021, 10, 3233.)

  1. Authors have mentioned that the results of the biopsies from enucleated eyes among patients who died COVID-19 shows the expression of spike protein and nucleocapsid in the retina. Is this expression found only in the severely infected patients or is it also predicted in mild to moderate infected people?

Answer to the review, not included in the text:

Retinal and choroidal biopsies are not performed among COVID-19 survivors, it is impossible to determine these viral proteins' levels in such patients and further correlate them with potential ocular consequences.

  1. The presence of cotton wool spots and microhemorrhages along the retinal vessels in adult patients with COVID-19 is present in all the types of patients or severely infected patients only? Provide this information more in detail. 

Answer to the review, not included in the text:

According to the Reviewer’s suggestion, we added more information on this subject.

Additional information included in the manuscript:

Section: Retinal ischemia; p. 4 of 14; line 599-602:

All these patients had fever, weakness, and dyspnoea, and 11 had no sense of smell. Two patients were admitted to the hospital; the rest were treated on an outpatient basis. No patient required intensive care. All patients had normal blood parameters on ophthalmological examination.

Section: Retinal ischemia; p. 4 of 14; line 607-610:

The exact physiopathology of cotton wool spots after COVID-19 is not known. Probably the vascular obstruction caused by direct viral tissue injury and infiltration of the endothelial cells via the ACE2 receptor can lead to this symptom [36].

  1. The study reports on reduced vessel density in patients shows both significant and no statistically significant reduction in VD in CC, and authors have reported that a number of factors may influence the difference in parameters, if authors discuss these factors more in detail it could be more informative.

Additional information included in the manuscript:

Section: Reduced Vessel Density in Patients with COVID-19 ; p. 6 of 14; line 1235-1241:

The above studies show discrepancies in statistical significance for the VD parameter in COVID-19 patients. This may be due to the small study groups, to differences in the time of the first eye examination from the end of hospitalization. Some were examined for the first time 14 days, while others were examined 12 weeks after leaving the hospital [42,43,46,47]. There are no precise data on the duration of hospitalization for these patients. In most cases, patients were not examined ophthalmologically during their hospitalization due to the risk of staff being infected with SARS-CoV-2.

  1. Authors have mentioned that hypoxia is responsible for the increased FAZ in DCP in patients, but also it was reported that no difference between  

Answer to the review, not included in the text:

As suggested, we added more information.

Additional information included in the manuscript:

Section: The Enlarged Foveal Avascular Zone In Patients With COVID-19; p. 6 of 14; line 1267-1271.

Differences in the FAZ parameter between COVID-19 patients and control groups may be explained by factors similar to those that may impinge on the VD parameter mentioned above. Perhaps the patients' age, the burden of additional diseases, saturation, treatment during hospitalization, and the time of the first ocular examination after hospital discharge.

  1. Authors have mentioned that the changes in retinal and choroidal microvascular parameters on OCTA were still 230 found 6 months later, is this report found in only severely infected patients, or in both mild and moderate infection?

Answer to the review, not included in the text:

It is found not only in severely infected patients. As suggested, we have written it more directly.

Additional information included in the manuscript:

Section: Persistent Reduced VD and Enlarged FAZ at 6-Months Follow-Up; p. 7; line: 1644:

These patients did not require a stay in the intensive care unit.

Section: Persistent Reduced VD and Enlarged FAZ at 6-Months Follow-Up; p. 7; Line: 1653-1655:

None of the patients in this study required a stay in the intensive care unit; 26 patients of the 63 hospitalized had confirmed bilateral pneumonia by computerized tomography scans.

Reviewer 2 Report

“The reduced vessel density and enlarged foveal avascular zone in the macula as a result of systemic hypoxia caused by SARS-CoV-2 infection” is informative manuscript. The manuscript needs revision before publication. I have the following suggestions for authors to address.

1.     Check the abbreviations throughout the manuscript and introduce the abbreviation when the full word appears the first time in the text and then use only the abbreviation (For example, OCT, VD, FAZ, ACE2). Make a word abbreviated in the article that is repeated at least two times in the text, not all words to be abbreviated (For example, GCL).

2.     There is a lack of recent literature citations. For example, in lines 30-32 “Moreover, a wide range of persistent symptoms can remain after SARS-CoV-2 infection, known as post-COVID-19 syndrome (post-coronavirus disease 2019), which can encompass sensory, neurologic, cardiorespiratory systems, and mental health. (DOI: 10.3389/fpubh.2022.908757)”.

3.     To better understand “the changes in retinal and choroidal microvasculature in individuals who underwent SARS-CoV-2 and its variants”, the authors should add “case reports published between January 2020 and January 2023”.

4.     In “Conclusions” section, the conclusion seems very general, lacking the future aspects and major findings. The quality of the conclusion should be improved.

Editing of English language required

Author Response

Reviewer 2

  1. Check the abbreviations throughout the manuscript and introduce the abbreviation when the full word appears the first time in the text and then use only the abbreviation (For example, OCT, VD, FAZ, ACE2). Make a word abbreviated in the article that is repeated at least two times in the text, not all words to be abbreviated (For example, GCL).

Answer to the review, not included in the text:

Thank you for that excellent and insightful series of remarks. We have changed the abbreviations.

  1. There is a lack of recent literature citations. For example, in lines 30-32 “Moreover, a wide range of persistent symptoms can remain after SARS-CoV-2 infection, known as post-COVID-19 syndrome (post-coronavirus disease 2019), which can encompass sensory, neurologic, cardiorespiratory systems, and mental health. (DOI: 10.3389/fpubh.2022.908757)”.

Answer to the review, not included in the text:

According to the Reviewer’s suggestion, we have added recent literature citations. It is mentioned below.

  1. To better understand “the changes in retinal and choroidal microvasculature in individuals who underwent SARS-CoV-2 and its variants”, the authors should add “case reports published between January 2020 and January 2023”.

Answer to the review, not included in the text:

According to the Reviewer’s suggestion, we have added case reports published between January 2020 and January 2023.

Additional information included in the manuscript:

Section: Future Research Prospects; p. 9 of 14; line 2282-2291.

Whether the effect of SARS-CoV-2 infection on the microcirculation of the retina and choroid could differ between different viral variants is not well explored. One should note that such a phenomenon may be expected since there are significant differences in the clinical course of COVID-19 between particular lineages, with a highly transmissible Omicron variant recognized as the least severe [85–88]. Moreover, particular subvariants of Omicron reveal differences in fusogenicity, which is known to affect viral pathogenicity [89]. The potential effects of infection may also vary in relation to viral loads [90,91]. Whether vaccination status may attenuate the effect of SARS-CoV-2 on the retinal and choroidal vasculature of the eye during breakthrough infection also remains to be explored.

There are also case reports of COVID-19 patients diagnosed with fundus lesions. One study described a 46-year-old man with no additional disease and saturation on the day of hospital admission of 93%. Three days after initial respiratory symptoms, fundus examination showed cotton wool spots and tortuous retinal vessels. OCT examination showed a thickening of the inner layers of the retina. 3 months later, all vascular changes present on the fundus had resolved. The visual acuity was normal during both ocular examinations [37].

Both central retinal vein and artery occlusions have been reported in patients treated due to COVID-19 who have no typical systemic vascular risk factors. Plausibly the complement-induced prothrombotic and inflammatory state induced by the virus is responsible for the endothelial damage and microangiopathic injury [38]. Yahalomi et al.described a previously healthy 33-year-old female with central retinal vein occlusion with COVID-19 [39]. Invernizzi et al. found retinal haemorrhages, cotton wool spots, dilated veins, and tortuous vessels in 54 patients with COVID-19 patients in fundus examination. As suggested, retinal vein diameter correlated directly with disease severity. This symptom can be a non-invasive parameter to monitor inflammatory response and endothelial injury caused by SARS-CoV-2 infection [40].

  1. In “Conclusions” section, the conclusion seems very general, lacking the future aspects and major findings. The quality of the conclusion should be improved.

Answer to the review, not included in the text:

According to the Reviewer’s suggestion, we improved the conclusion.

Additional information included in the manuscript:

Section: conclusions; p. 10 of 14; line 2310-2312:

Due to the history of COVID-19 disease, a general medical history of SARS-CoV-2 infection should be considered in the differentiation of macular disease based on OCTA examination.

Round 2

Reviewer 1 Report

Revised version can be accepted for publications